# Experimental investigation of the creep damage evolution of coal rock around gas extraction boreholes at different water contents

Jiangbo Guo[1,2], Tianjun Zhang[1,2]*, Hongyu Pan[1,2]*, Jinyu Wu[1,2]

1 College of Safety Science and Engineering, Xi'an University of Science and Technology, Xi'an, China,
2 Key Laboratory of Western Mine Exploitation and Hazard Prevention of the Ministry of Education, Xi'an, China

* tianjun_zhang@xust.edu.cn (TZ); panhy@xust.edu.cn (HP)

## Abstract

The creep process of the coal rock around the extraction boreholes under stress-water coupling is an important factor affecting the stability of the boreholes. To study the influence of the water content of perimeter of the coal rock around the boreholes on its creep damage, a creep intrinsic model considering water damage was established by introducing the plastic element model from the Nishihara model. To study the steady-state strain and damage evolution of coal rocks containing pores, and verify the practicality of the model, a graded loading water-bearing creep test was designed to explore the role of different water-bearing conditions in the creep process. The following conclusions were obtained: 1) water has a physical erosion and softening water wedge effect on the perimeter of the coal rock around the boreholes, which affects the loading axial strain and displacement of the perforated specimens; 2) an increase in water content reduces the time taken for perforated specimens to enter the creep phase, making the accelerated creep phase come earlier; 3) the parameters of the water damage model are considered to be exponentially related with the water content. The experimental data are similar to the results of the model parameters, and the model shows some practicality; 4) the damage variables in the accelerated creep phase increase rapidly throughout the creep process, leading to local instability in the borehole. The findings of the study provide important theoretical implications for the study of instability in gas extraction boreholes.

## Introduction

Creep damage to coal rock around gas extraction boreholes is an important factor threatening the stability of the boreholes [1, 2]. The creep of the coal rock around the boreholes caused by the stress-water solid-liquid coupling results in a variable and complex fracture extension process [3]. To ensure the safety of coal seam mining and to improve the efficiency of gas extraction, the phenomenon of coal rock creep must be studied. Therefore, it is necessary to estimate

**Funding:** This work was supported by the National Natural Science Foundations of China, Study on mechanism and parameter optimization of carbon dioxide deep hole pre-split blasting in coal seam (Grant No. 51874234). The Funder provides an important role for the paper; The main research idea and manuscript preparation were contributed by Jiangbo Guo; Tianjun Zhang contributed on the manuscript preparation and performed the correlative experiment. Hongyu Pan gave several suggestions from the industrial perspectives. Jinyu Wu assisted on finalizing research work and manuscript.

**Competing interests:** The authors declare that there is no conflict of interest regarding the publication of this paper.

the effects of different water contents on the creep process of coal rocks, starting from the creep intrinsic model, and to establish a creep intrinsic model considering water damage; through the damage structure evolution of the coal rock containing the hole, thus provide a theoretical basis for the management of borehole destabilisation damage.

In recent years, research into the creep properties of coal rock bodies has been conducted, with the focus being on empirical and component models. For pore-bearing coal bodies under different external conditions, the empirical models obtained through experiments are inconsistent, but in general most cases use power functions, exponential functions, logarithmic functions, and hybrid forms of creep models [4, 5]. Aubenin et al. [6] obtained the creep properties of a certain structural surface through the study of empirical model equations. Sun et al. and Yan et al. [7, 8] conducted creep tests with different perimeter pressures under triaxial stress conditions and used the empirical creep equations to conduct an in-depth analysis of deformation damage of clay under normal conditions, the creep intrinsic model, and the correlation of parameters in the model. Shi et al. [9] conducted uniaxial compression creep tests under different axial stresses, obtained creep curve-based test results, analysed the mechanical properties of specimens with different prefabricated crack lengths and investigated the applicability of traditional empirical equations to creep damage characteristics. Hu et al. [10] conducted graded loading creep experiments and established a non-linear creep damage intrinsic model and equations for rock materials, describing the strength damage evolution of the rock during the creep damage process. Wang et al. [11] developed a creep instantiation model based on the results of uniaxial compression creep tests on salt rocks and triaxial compression creep tests on coal materials to reflect the entire creep process. Ma et al. [12] proposed a new quadratic creep instantiation model based on variable order fractional derivatives and continuum damage mechanics to allow the creep model to describe the long-term and short-term damage properties of such materials.

Regarding the influence of water-bearing coal rock masses on their creep damage properties, Hashiba et al. [13] conducted tests on the creep properties of materials in the water-bearing state and obtained a relationship between the creep strain and the water content. Lin et al. and Mendes et al. [14, 15] conducted water-bearing creep tests on rock materials and described the relationship between water content and stress on the creep strain, creep rate, long-term strength, and creep damage mode. Yu et al. [16] realised the creep damage of water-bearing rocks through a rock creep intrinsic structure relationship, and the coupling of creep damage, and verified the effect of the water content on rock damage. Lui et al. [17] studied the creep properties of limestone under fluid-solid coupling and discussed the effects of water and hydraulic envelope pressure on the damage mechanism and creep properties. Yan et al. [18] developed a triaxial creep test for water-bearing materials to determine the effect of water-force coupling on the creep of rocks, and proposed a non-linear empirical model of creep instantiation by considering the creep damage mechanism. Jia et al. [19] investigated the creep pattern of mud rocks under the action of water-force coupling, and built a non-linear creep instantiation model considering water-force coupling by establishing the relationship between creep damage and creep strain. Wang et al. [20] and Zhang et al. [21] through the analysis of the creep damage characteristics of coal rock, the creep mutation model of coal rock damage is proposed, and its intrinsic equation and the time of creep damage of coal rock body when the stress reaches a certain level are derived.

However, few of the existing models consider the influences of different factors on the damage characteristics of the boreholes. The better to reflect the damage pattern around a given borehole, the authors start from the creep model and establish a creep intrinsic structure model considering different damage factors. The evolution of the model parameters considering water-contaminated ageing damage was also obtained through laboratory graded loading

creep tests. The relationship between the model parameters and the experimental data was compared to verify the practicality of the creep damage intrinsic model for coal rock around the water-bearing gas extraction boreholes. The result provides an important theoretical basis for the prediction and warning of gas extraction borehole instability accidents and support work.

## Creep intrinsic model considering damage

During prolonged gas extraction, complex stress distribution can cause creep in the coal rock around the boreholes; the presence of water during the drilling process makes the coal rock creep process variable and complex. Creep damage to coal rock readily induces deformation of extraction boreholes resulting in defects such as collapsed boreholes and instability, which can lead to inefficient gas extraction. Therefore, investigations of the creep damage nature of the coal rock around the extraction boreholes and the creep pattern of the coal body around the boreholes under the joint action of stress and water can further improve the research of the damage mechanism of the extraction boreholes, which is of great significance to the study of the stability of the boreholes.

### Improved Nishihara model and principal structure equations

Creep in the coal rock around boreholes is a phenomenon where a certain stress is applied to the coal rock material and remains constant, and the strain increases with time [22]. The creep damage model simulates the creep process of materials from various aspects by connecting elastic, viscous, and plastic elements in series and parallel, which can reveal the underlying creep mechanism [23, 24].

The creep of coal rock around the boreholes can be divided into two main categories: one is at lower stresses, where the coal rock material goes through a deceleration creep phase and then eventually remains in a stable creep phase; the other pertains to higher stresses, where the creep process of the coal rock material goes through three phases: decelerating creep, stable creep, and accelerating creep [25].

In general, the coal rock material around the perimeter of the extraction boreholes is under a high stress. To reflect the whole creep process, the model established needs to be able to show the mechanical characteristics of the elastic strain phase, decelerating creep phase, stabilisation creep phase, and accelerating creep phase. Most scholars describe the creep process of coal rock materials through the classic Nishihara model (Fig 1).

In Fig 1: I is the Hooke body; II is the Kelvin body; III is the visco-plastic body; $\sigma$ is the stress applied to the material; $\sigma_s$ is the stress at which the material is first damaged; $E_0$ and $E_1$ refer to the modulus of elasticity of the Hooke body and Kelvin body respectively; $\eta_1$ and $\eta_2$ represent the viscous coefficients of the Kelvin body and visco-plastic body respectively.

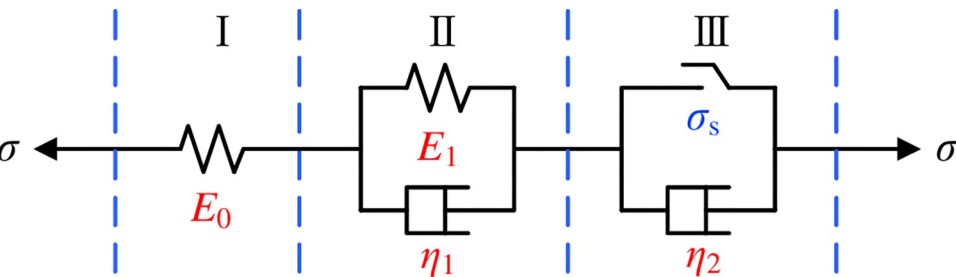

**Fig 1. The classic Nishihara model.**

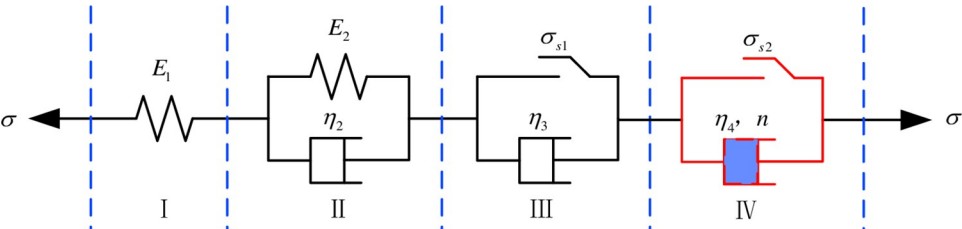

**Fig 2. Improved Nishihara model considering accelerated creep cells.**

The Nishihara model mainly reflects the mechanical properties of the material in the decelerating creep and steady creep stages, but not in the accelerated creep stage of the specimens containing pores. Therefore, this paper improves the Nishihara model by adding the NVPE visco-plastic model to establish a damage creep model under the influences of different factors. The improved Nishihara model considering accelerated creep cells is shown in Fig 2.

Where: $E_1$ and $E_2$ denote the modulus of elasticity of the specimen; $\eta_2$, $\eta_3$ and $\eta_4$ refer to the coefficient of viscosity; $\sigma_{s1}$ and $\sigma_{s2}$ represent the long-term strength and yield strength of the specimen respectively; and $\sigma_{s1}<\sigma_{s2}$, $n$ is the creep coefficient of the specimen; I is a description of the deformation process of the coal rock material when loaded, II is a description of the process of the coal rock material into the decelerating creep stage, III is a description of the process of the coal rock material into the stable creep stage, and IV is a Bingham body, revealing the process of the coal rock material into the accelerated creep stage.

The creep equation can be obtained as follows:

$$\varepsilon = \frac{\sigma}{E_1} + \frac{\sigma}{E_2}\left(1 - e^{-t \cdot E_2/\eta_2}\right) + \frac{\sigma - \sigma_{s1}}{\eta_3} t + \frac{\sigma - \sigma_{s2}}{\eta_4} t^n \tag{1}$$

The one-dimensional creep equation for the modified West Plains model is then obtained as:

$$\varepsilon = \begin{cases} \dfrac{\sigma}{E_1} + \dfrac{\sigma}{E_2}\left(1 - e^{-t \cdot E_2/\eta_2}\right) & \sigma < \sigma_{s1} \\[2mm] \dfrac{\sigma}{E_1} + \dfrac{\sigma}{E_2}\left(1 - e^{-t \cdot E_2/\eta_2}\right) + \dfrac{\sigma - \sigma_{s1}}{\eta_3} t & \sigma_{s1} \leq \sigma < \sigma_{s2} \\[2mm] \dfrac{\sigma}{E_1} + \dfrac{\sigma}{E_2}\left(1 - e^{-t \cdot E_2/\eta_2}\right) + \dfrac{\sigma - \sigma_{s1}}{\eta_3} t + \dfrac{\sigma - \sigma_{s2}}{\eta_4} t^n & \sigma_{s2} \leq \sigma \end{cases} \tag{2}$$

From the above equation, the modified Nishihara creep model curve before and after the long-term strength is mobilised can be divided into three types of variation: (1) When the loading stress is less than the long-term strength, the creep rate of the stable creep phase is 0; (2) When the loading stress is higher than the long-term strength and lower than the breaking strength, the stable creep rate of the material refers to the ratio of the loading stress minus the long-term strength stress to the viscosity coefficient in the Bingham body; (3) When the loading stress first exceeds the yield stress, the accelerated creep phase begins; over time, the creep deformation rate gradually increases and the creep characteristics become more significant.

## Creep intrinsic model considering water damage

Much water is usually present during the extraction of gas from underground wells. Water has a softening, mudification effect on the coal rock, which affects the creep process of the coal rock and aggravates the instability and deformation damage of the boreholes. The creep

intrinsic model reflects the destabilisation of the coal rock around the boreholes over a long period of time and focuses less on the influences of other factors affecting the damage to the boreholes. Therefore, the better to reflect the damage around a borehole, this section introduces the plastic element model based on the traditional model and establishes a creep principal structure model considering different water damage factors.

Stress state, loading time and water content all have a significant impact on the creep characteristics of the coal rock around the perimeter of gas extraction boreholes. Therefore, in conjunction with Eq (2), an instantaneous creep model that takes into account the water content damage $D_w$ and the ageing damage $D_t$ is developed. According to the elastic modulus damage theory, the damage variables considering water content and time can be determined as:

$$D_w = 1 - \frac{E_w}{E_0}, \ D_t = 1 - \frac{E_t}{E_0} \tag{3}$$

Where: $D_w$ is the water damage variable; $D_t$ denotes the time damage variable; $E_w$ is the modulus of elasticity of the water damage variable; $E_0$ represents the modulus of elasticity of the undamaged state (assuming zero water content); $E_t$ is the modulus of elasticity at any point in time.

Therefore, by combining Eqs (2) and (3), the creep intrinsic equation that considers water damage and time damage is obtained:

(1) When $\sigma < \sigma_{s1}$,

$$\varepsilon = \frac{\sigma}{E_1(1 - D_w)(1 - D_t)} + \frac{\sigma}{E_2(1 - D_w)(1 - D_t)}\left(1 - \exp(-\frac{E_2(1 - D_w)}{\eta_2})t\right) \tag{4}$$

(2) When $\sigma_{s1} \leq \sigma < \sigma_{s2}$,

$$\varepsilon = \frac{\sigma}{E_1(1 - D_w)(1 - D_t)} + \frac{\sigma}{E_2(1 - D_w)(1 - D_t)}\left(1 - \exp(-\frac{E_2(1 - D_w)}{\eta_2})t\right)$$
$$+ \frac{\sigma - \sigma_{s1}}{\eta_3(1 - D_t)}t \tag{5}$$

(3) When $\sigma_{s2} < \sigma$,

$$\varepsilon = \frac{\sigma}{E_1(1 - D_w)(1 - D_t)} + \frac{\sigma}{E_2(1 - D_w)(1 - D_t)}\left(1 - \exp(-\frac{E_2(1 - D_w)}{\eta_2})t\right) +$$
$$\frac{\sigma - \sigma_{s1}}{\eta_3(1 - D_t)}t + \frac{\sigma - \sigma_{s2}}{\eta_4(1 - D_t)}t^n \tag{6}$$

Where, in Eq (2), $\eta_2$, $\eta_3$ and $\eta_4$ are the coefficient of viscosity with respect to the loading time; $E_1$ and $E_2$ denote the modulus of elasticity with respect to the water content and time.

## Materials and methods

The creep properties of the coal rock around the boreholes are of practical importance to the stability of the boreholes. Long-term stresses and the influence of water can cause creep damage to the coal rock. Therefore, in this paper, to study the influence of the water-bearing coal rock around the boreholes on its creep evolution process, a creep test system designed by

ourselves was used to determine the creep characteristic parameters of the hole-containing specimen, and then its creep intrinsic model was improved, which can better reflect the instability-related damage to the coal rock around the boreholes under the long-term stress-water coupling effect.

## Materials

(1) **Research background.** The Yuwu Coal Company wellfield is situated in Tunliu and Xiangqi counties in Shanxi Province, to the west of Lu'an Mining (Group) Company. The Yuwu coalfield lies in the western part of the middle section of the Taihang Mountains, west of the Changzhi Basin and in the middle of the eastern part of the Qinshui coalfield. Yuwu Coal Company mines the 3# coal seam, with an inclination of 3 to 15˚, thickness 5.0 to 7.25 m, and an average thickness of 5.99 m. The old top is sandy mudstone, with a thickness of 9.78 m. The direct top is mudstone, with a thickness of 2.25 m. The direct bottom is siltstone and fine-grained sandstone, with a thickness of 6.10 m. The old bottom is sandy mudstone, with a thickness of 8.4 m. The working face adopts the integrated coal mining method of releasing the top coal, the coal tunnelling working face adopts integrated mining, the annual production capacity of the mine reaches 8 million tons. The 3# coal seam gas content is 3.06–23.69 $m^3$/t, the average gas content is 8.51 $m^3$/t, the residual gas content is 2.37 $m^3$/t, the gas pressure is 0.42–0.87 MPa, the total amount of gas is about 16569 $Mm^3$. The solidity coefficient of the coal body is 0.44–0.53 (a loose coal seam), the permeability is 0.5240–1.7415 $m^2$/($MPa^2$·d), the 3# coal seam can be classified as a high-gas, low-permeability, loose coal seam. The coal seam is in a location with many faults and a complex geological structure. Due to the influence of the trap column, geological changes such as associated faults, a broken roof and coal body, fissure development, gas anomalies and water gushing anomalies occur during the excavation process.

(2) **Material preparation.** Under the coupling of stress and water, the coal rock body around the perimeter of the gas extraction boreholes becomes more susceptible to damage compared to the intact coal rock body. This is due to the redistribution of stresses in the perforated coal body under the influence of the boreholes, which can create new areas of stress concentration. Typically, the diameter of the extraction boreholes is much smaller than the depth and it can be assumed (by way of an approximation) that the peri-hole stresses do not change with increasing length of the boreholes and that the stress boundary is three times the radius of the boreholes. Therefore, herein, rectangular specimens with boreholes were produced with a length, width and height of 70 mm, 70 mm, and 140 mm respectively, and a radius of 5 mm were drilled. During the fabrication of the specimens, the centre of the rectangular mould box was prefabricated with boreholes drilled, and then similarly proportioned specimens were used to simulate the coal rock around the boreholes, with the ratio of coal dust: cement, mixed with a mass ratio of 1:0.7 to make a slurry to fill the prefabricated mould; the prepared specimens were kept moist (to prevent air-drying) and in a cool place for 30 days. A total of 24 rectangular specimen moulds containing holes were prepared for this test, as shown in Fig 3.

Before conducting the test, the naturally air-dried specimens were first divided into four groups A, B, C, and D, and the initial mass of the specimens was tested. Next, the three groups B, C, and D were sealed in a water container and the liquid level in the container was controlled to be more than 100 mm away from the top surface of the specimen. After the container has been placed in the shade for 24 hours, the surface of the specimen was wiped until there were no more water droplets on the surface and weighed. This procedure was repeated every 24 hours until the mass of the specimen no longer changed, and the mass of the specimen was recorded. Finally, the specimens in group D were wrapped completely in plastic film and

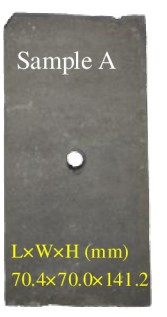 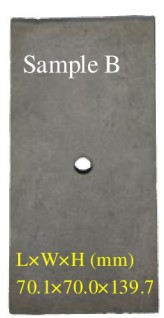 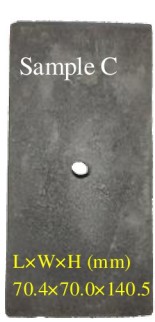 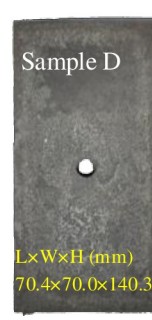

| | | | |
|---|---|---|---|
| **Sample A** | **Sample B** | **Sample C** | **Sample D** |

$\varepsilon=0.0\%$
$\omega=0.0124$
L×W×H (mm) $E=0.42$GP
70.4×70.0×141.2 $F=21.4$KN

$\varepsilon=10.4\%$
$\omega=0.0138$
L×W×H (mm) $E=0.37$GP
70.1×70.0×139.7 $F=17.62$KN

$\varepsilon=19.3\%$
$\omega=0.0149$
L×W×H (mm) $E=0.28$GP
70.4×70.0×140.5 $F=15.5$KN

$\varepsilon=28.4\%$
$\omega=0.0189$
L×W×H (mm) $E=0.18$GP
70.4×70.0×140.3 $F=13.8$KN

**Fig 3. Specimen preparation.**

stored. Groups B and C were oven-dried at 100°C. The specimens were each removed at 1-hour intervals and weighed when cooled to room temperature and the data were recorded until the water content of the specimens reached the required value; the specimens were then wrapped in plastic film for storage. The water content of the specimen can be calculated using the following formula:

$$w = \frac{m_1 - m_0}{m_0} \times 100\% \tag{7}$$

where: $w$ is the water content of the specimen prepared for the test, %; $m_0$ is the initial mass of the specimen, g; $m_1$ denotes the mass of the specimen with water, g.

## Methodologies

**(1) Experiment design.** According to the actual situation of the underground gas extraction site, when the underground working face boreholes are injected with water, the coal seam is usually in a completely submerged environment, and the water content of the coal rock around the boreholes is naturally full of water at this time, and as the distance from the boreholes increases, the water content of the coal rock gradually decreases, and finally the coal rock around the boreholes is in a natural water state. Therefore, to study the influence of the water content on the creep evolution of the coal rock around the boreholes, four groups of specimens, namely dry group A, controlled water content groups B and C, and natural saturation group D, with six specimens each, were set in this paper to meet the natural water content of the coal seam and the influence of external water injection thereon.

The 24 specimens containing holes made in this test, Ai, Bi, Ci and Di (i = 1, 2, or 3) were used for the uniaxial compressive damage tests and Ai, Bi, Ci and Di (i = 4, 5, or 6) were used for the graded creep loading test specimens

**(2) Experimental procedure.** This test was conducted in a graded loading mode because graded loading allows the loading at different stresses and the displacement changes during the constant load phase to be observed on the same specimen. Based on the uniaxial compressive damage test results for the four groups of specimens containing holes Ai, Bi, Ci, and Di (i = 1, 2, or 3), 50%, 60%, 70%, 80%, and 90% of their peak stresses were taken as the five stresses applied in the creep test loading; the stress applied at each level of the graded loading in the coal rock creep test was calculated according to the uniaxial compression test results (Table 1).

**(3) Experimental equipment.** The graded loading creep test system for perforated coal rock is mainly composed of the YYL200 electronic persistent creep testing machine, load bearing device and computer data acquisition system (Fig 4). The loading range of the tester is 0 to

Table 1. Loading parameters for graded load application.

| Specimen (Water content /%) | Stress levels at graded loading levels/ MPa | | | | |
|---|---|---|---|---|---|
| | First level | Second level | Third level | Fourth level | Fifth level |
| Sample A (0.0%) | 2.17 | 2.61 | 3.04 | 3.48 | 3.91 |
| Sample B (10.4%) | 1.74 | 2.09 | 2.43 | 2.78 | 3.13 |
| Sample C (19.3%) | 1.6 | 1.92 | 2.24 | 2.56 | 2.88 |
| Sample D (28.4%) | 1.38 | 1.66 | 1.93 | 2.21 | 2.49 |

200 kN and the loading rate is 0.01 to 80 mm/min. The test is set for a constant loading time of 120 min and a loading time of 10 min.

## Results and discussion

To study the effect of creep on the coal rock around the perimeter of gas extraction boreholes during long periods of stress-water coupling, the authors proposes a creep intrinsic model to study the effects of different water states on the creep deformation and steady-state strain of the hole-containing specimens, and validates the creep-damage model of coal rock with water aging based on the above study, and verifies the accuracy and reasonableness of the model through experimental data, and derives the damage evolution law of the coal rock containing a pre-drilled hole.

### Effect of the water content on creep processes in pore-bearing coal rocks

According to the graded loading test scheme, the graded creep tests of borehole specimens with different water contents were conducted at room temperature (35–40˚C). Since dispersion in the specimen preparation process is inevitable and will affect the experimental results, the axial strain curves of borehole-containing specimens with different water contents were obtained after averaging each group of specimen data (Fig 5).

As can be seen from the axial strain curves, at low stresses, the creep rate of the specimens containing holes first varied slowly, then tended to stabilise. As the stress was increased, steady and accelerated creep occurred in the specimens. This test reflects the basic process of creep in coal rocks and can thus be used to analyse the creep properties of the pore-bearing specimens.

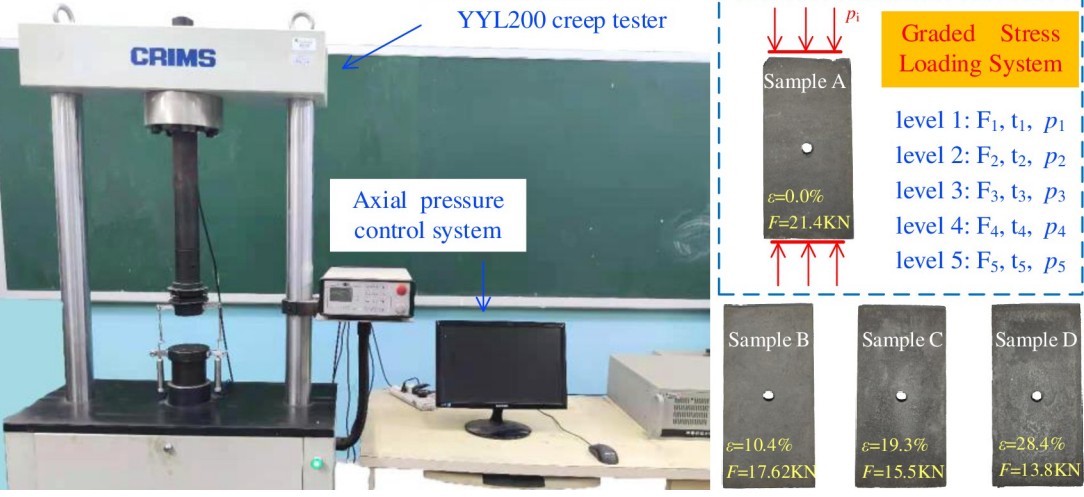

Fig 4. Diagram showing the test set-up.

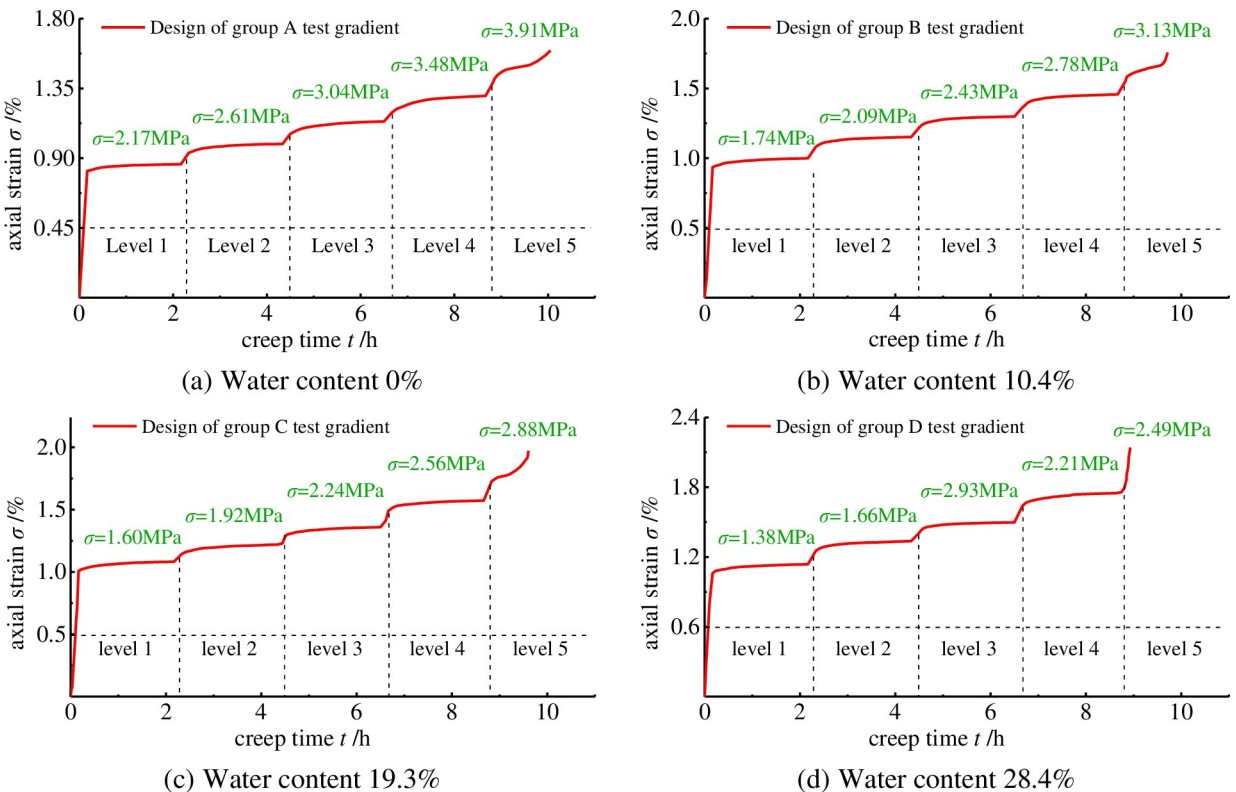

**Fig 5. Axial strain curves at different water contents.** (a) Water content 0%, (b) Water content 10.4%, (c) Water content 19.3%, (d) Water content 28.4%.

At the same water content, the creep curve at 0% water content, for example, produced the smallest change in strain-deformation at each level compared to the specimens at other water contents, and the creep process and ultimate strain process were slow until damage occurred. At the same stress level, the compressive strength, creep time and ultimate strain all tended to decrease as the water content of the specimen increased, with only the instantaneous strain increasing. Furthermore, the increase in water content causes a small increase in the instantaneous axial deformation and creep rate in the steady state at the same applied stress, while the modulus of elasticity and viscous modulus of the specimen gradually decrease during the loading process. The creep curves of the porous specimens show an overall stepped upwards at different water contents. With the gradual increase of the applied stress, the creep curves exhibited three stages, respectively: (1) decelerating creep; (2) stable creep; (3) accelerating creep. With the increase of the water content, the axial strain of the specimens gradually increased: the specimens with 0% water content entered the accelerated creep stage at an axial strain of 1.49%; the specimens with the water content of 10.4% entered into the accelerated creep stage at an axial strain of 1.65%; the specimens with the water content of 19.3% entered the accelerated creep stage at an axial strain of 1.74%; the specimen with 28.4% water content entered the accelerated creep stage at an axial strain of 1.8%.

From the creep curves of specimens with holes in different water content states, Fig 6 shows that the time for the specimens to enter the stable creep stage is within 50–70 min at each applied stress. With the increase of the water content, the creep damage deformation of the pore-containing specimens under each applied stress gradually increased, which accelerated the rate of the creep phase of the specimens and made the accelerated creep phase appear

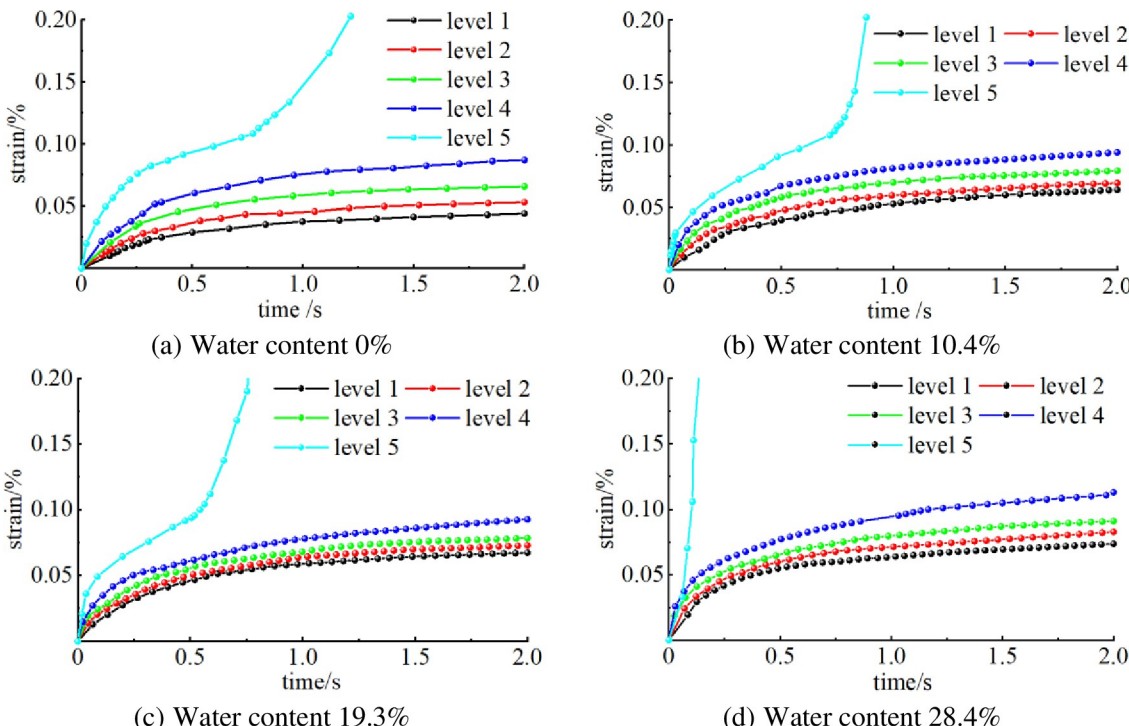

**Fig 6. Creep curves of specimens with pores at different water contents.** (a) Water content 0%, (b) Water content 10.4%, (c) Water content 19.3%, (d) Water content 28.4%.

quicker. The creep phase developed more rapidly at the end of the fifth stress level among the five applied stresses. At the fifth stress level, the specimen with a water content of 28.4% did not experience the deceleration and stabilisation of the creep phase and entered the accelerated creep phase directly.

Comparison of the creep characteristics curves of the specimens with four different water contents shows that the pores, the maximum creep deformation and creep deformation gradually increased with increasing water content. The maximum axial strain in the accelerated creep phase of the porous specimens was increased by 0.31% for the 28.4% water content specimens compared to the 0% water content specimens. This finding reflects the physical erosion and softening effect of water on the porous specimens. On a microscopic level, water can wet the particles on the free surface of the coal body, thus reducing the cohesion between coal particles and affecting the mechanical properties of the overall coal body, making the stressed perimeter coal body more susceptible to creep deformation, so water can have a greater effect on the loaded axial strain and displacement of the perforated coal sample.

## Steady-state strain during creep in pore-bearing coal rocks

The steady-state strain $\varepsilon_w$ is the strain at which the perforated coal rock stabilises in creep deformation and this occurs at the intermediate stage of the creep process, namely the steady creep stage [26, 27]. In the modified West Plains model, which uses the viscoelastic body III to describe the process by which the coal rock material enters the steady-state creep phase, the steady-state strain is calculated as follows.

$$\varepsilon_w = \frac{\sigma - \sigma_s}{\eta_3} t \tag{8}$$

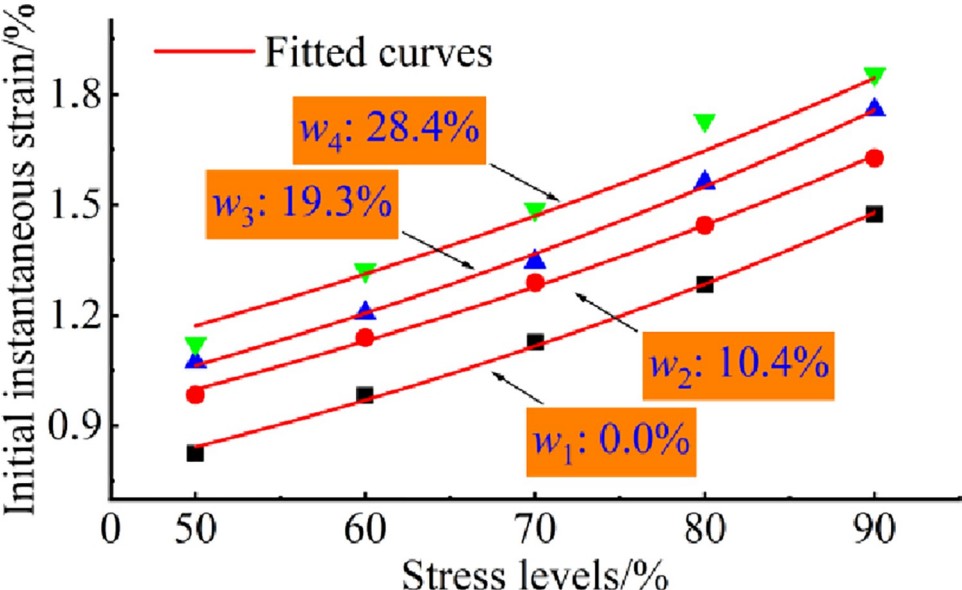

**Fig 7. Steady-state variation curves for specimens subject to different stresses.**

Where: $\sigma$ is the applied axial stress, $\sigma_s$ is the constant load strength of the specimen and $\eta_3$ is the coefficient of viscosity of the specimen in the steady creep stage.

The effects of stress and water on creep damage of coal rock materials can be reflected laterally, as shown in Figs 7 and 8.

The relationship between applied stress and steady-state strain is shown in Fig 7, at the same water content condition, the stable strain of the pore-containing specimen increases exponentially with increasing stress level and is directly related to the value of the water content. The correlation coefficients of the fitted equations for the stable strain and stress levels are all above 0.93 for, $0.417 \times e^{(0.0141\sigma)}$, $0.541 \times e^{(0.0123\sigma)}$, $0.568 \times e^{(0.0123\sigma)}$ and $0.665 \times e^{(0.0113\sigma)}$ respectively. At each applied stress, the steady-state strain compared to the 0% water content

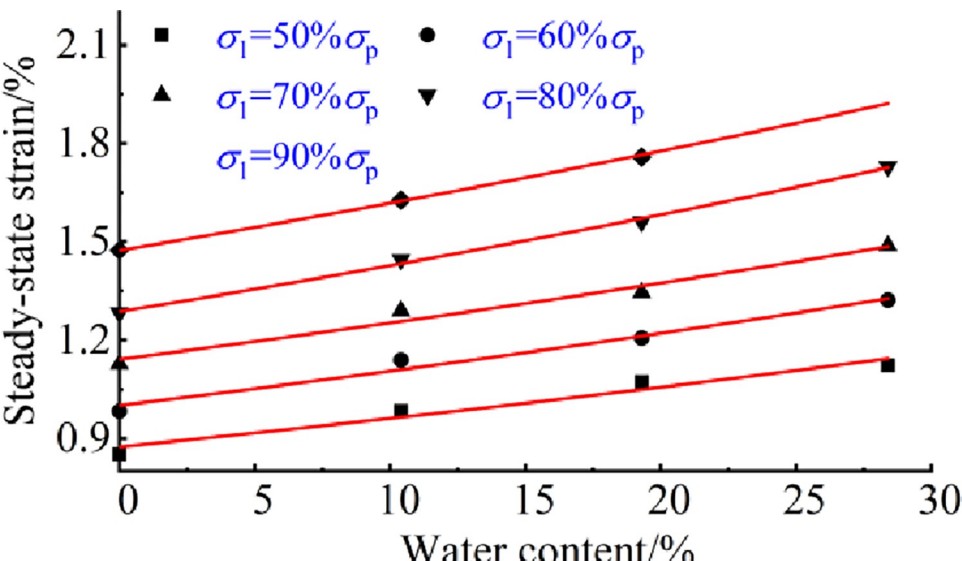

**Fig 8. Steady-state strain curves for specimens with different water contents.**

specimen is increased by 10–16% for the 10.4% water content specimen, increased by 19–22% for the 19.3% water content specimen and increased by 31–35% for the 28.4% water content specimen.

The relationship between water content and steady state strain is displayed in Fig 8, which shows that the steady state strain of the contained pore increases exponentially with increasing water content at the same applied stress. The fitted equations, in descending order of the water content, are $0.875 \times e^{(0.00945w)}$, $1.001 \times e^{(0.0993w)}$, $1.146 \times e^{(0.00927w)}$, $1.287 \times e^{(0.01033w)}$ and $1.474 \times e^{(0.00935w)}$, with correlation coefficients ranging from 0.956 to 0.997, which show a high correlation, indicating that water content has an effect on steady-state strain.

These findings indicate that the presence of water accelerates creep damage in perforated coal rocks. During uniform creep, the steady-state strain of the perforated coal body increases with increasing water content and is specifically related to the water content. The relationship between steady state strain and water content indicates that the ultimate creep deformation in the uniform creep phase of the perforated coal body increases with increasing water content, the more pronounced the creep deformation damage is in the naturally water-filled state. Therefore, the greater the water content of the coal rock around a borehole, the more severe the creep deformation, and the greater the probability of the borehole collapse due to instability under continuous stress. Therefore, the study of the effect of the water content on creep deformation of the perimeter of coal body can provide important guidance for gas extraction, drilling and support operations.

## Creep-damage model for coal rocks considering water-bearing ageing

This section combines the water-bearing time-dependent damage creep intrinsic model established in Section 2, processes the test data by using the least squares method, obtains the relevant parameters of the model and the regression effect, and summarises the variations of the parameters, analyses and compares the model calculation results with the test data, and verifies the accuracy and reasonableness of the model. In addition, the relationship between damage variables and water content of the specimen was adopted to study the creep damage characteristics of coal rock around the boreholes.

**(1) Parameters of the water-induced ageing damage model.** When using the water-contaminated ageing damage model, it is necessary to determine the two parameters required in the model: the long-term strength $\sigma_{s1}$ and the yield strength $\sigma_{s2}$. The long-term strength $\sigma_{s1}$ and yield strength $\sigma_{s2}$ are defined as the excess stress in the creep process from the decelerating creep stage to the stabilising creep of the coal rock material under a constant stress [28]. By analysing the data obtained from these tests, we define $\sigma_{s1}$ is the upper stress at which the last stable creep stage of the coal rock material occurs and $\sigma_{s2}$ is the upper stress at which the accelerated creep stage begins, therefore, the long-term strength $\sigma_{s1}$ is set to the stress at the third stress level and the yield strength $\sigma_{s2}$ is set to the stress at the fourth stress level.

Using the least-squares method, creep data for specimens containing pores at different water contents were fitted, where the fourth level of stress level strain data could be selected as the stable creep stage. The results of the fitting of the non-linear regression of the test data by least squares are illustrated in Fig 9, where the fitting coefficients are all above 0.98 suggesting that the correlation is high.

**(2) The variations of model parameters with the water content.** Based on the above data fitting study, the regression effects were averaged separately for specimens with different water contents to obtain the l parameter values of the creep mode (Table 2).

According to Table 2, the variation law of the model parameters with the water content can be obtained, then the fitting equation for each parameter of the model considering water-

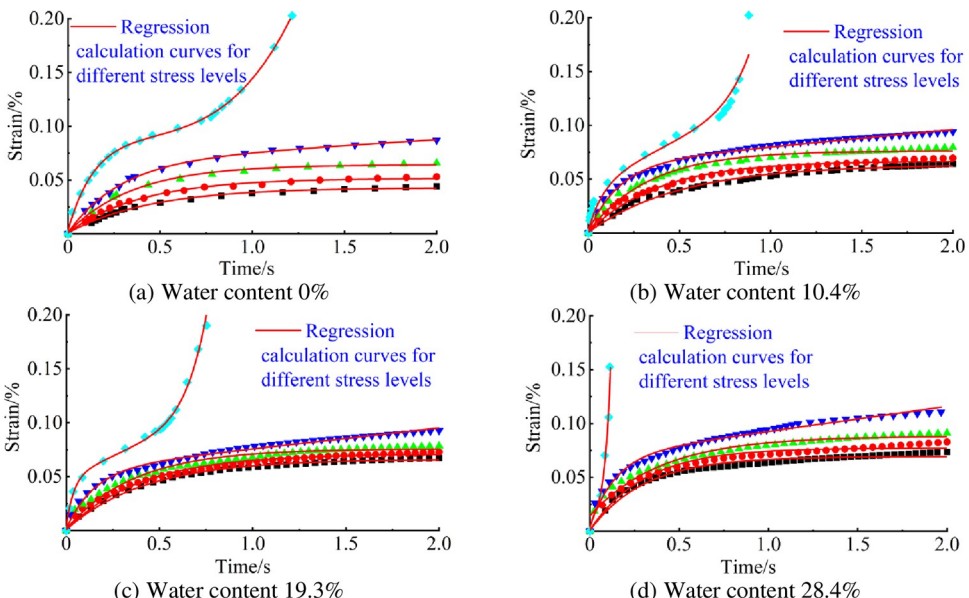

**Fig 9. Regression effects of the age-damaged creep model considering the water content.** (a) Water content 0%, (b) Water content 10.4%, (c) Water content 19.3%, (d) Water content 28.4%.

contaminated ageing damage is:

$$\left.\begin{array}{l} E_1 = 348e^{-0.02w}, \ E_2 = 64e^{-0.02w} \\ \eta_2 = 17e^{-0.02w}, \quad \eta_3 = 43e^{-0.05w} \\ \eta_4 = 10e^{-0.08w}, \quad n = 4e^{0.02w} \\ D_w = 0.12e^{0.02w}, \ D_t = 0.11e^{0.03w} \end{array}\right\} \tag{9}$$

According to Fig 9 and Eq (9), each parameter in the model considering water content damage has an exponential relationship with the water content, and the correlation coefficients are all above 0.95, suggesting a good correlation. The model parameters $E_1$, $E_2$, $\eta_2$, $\eta_3$, and $\eta_4$ decrease with increasing water content, and $n$, $D_w$, and $D_t$ increase with increasing water content.

**(3) Procedure for testing the water-bearing ageing creep model.** The fitted Eq (9) considering the parameters of the water-bearing ageing damage model was substituted into Eqs (4), (5), and (6) and the results obtained were compared with the creep curves of the test pore-containing specimens to plot the curves of the test values and the calculated values of the creep model under different water-bearing conditions (Fig 10).

Fig 10 indicates that the data calculated by the creep model under different water contents and the data obtained from the test are similar throughout the creep process, and although there are some errors, the overall trend is similar, indicating the accuracy and reasonableness of the creep intrinsic structure model considering the ageing damage of the water content.

**Table 2. Mean values of the parameters of the ageing damage model considering water content.**

| Specimen | Water content/% | $E_1$/ MPa | $E_2$/ MPa | $\eta_2$/(MPa·h) | $\eta_3$/(MPa·h) | $\eta_4$/(MPa·h) | $n$ |
|---|---|---|---|---|---|---|---|
| A | 0 | 338.67 | 62.68 | 18.13 | 44.18 | 10.53 | 4.25 |
| B | 10.4 | 298.35 | 57.96 | 12.24 | 21.57 | 4.64 | 5.28 |
| C | 19.3 | 224.56 | 45.65 | 9.79 | 16.66 | 2.33 | 6.52 |
| D | 28.4 | 164.95 | 41.28 | 8.33 | 11.62 | 1.19 | 7.28 |

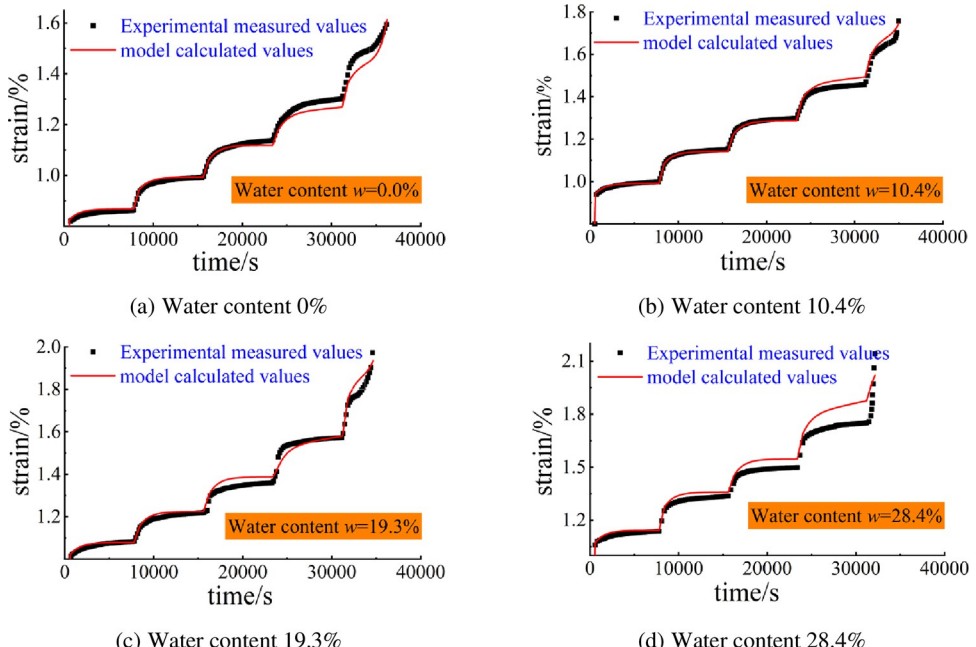

**Fig 10. Ageing creep model with water; comparison of calculated and measured values.** (a) Water content 0%, (b) Water content 10.4%, (c) Water content 19.3%, (d) Water content 28.4%.

## Structural evolution of damage in porous coal rocks

Under the action of long-term axial loading, the porous specimens were subjected to internal sprouting, expansion, evolution of microfractures, and finally formation of penetration cracks leading to damage. Based on the results of the graded loading creep experiments on porous specimens in this paper, it is found that there is a relationship between creep damage and creep strain in coal rock materials [29, 30]. According to the definition of the damage variables, the creep strain at each stage of the creep process is taken as the damage variable of the specimen, and assuming that the damage variable is 1 when the specimen is damaged and the coal rock is in a non-damaged state when the water content is 0%, the creep strain damage variable of the coal rock material is written as [31, 32].

$$D = \bar{\varepsilon}/\varepsilon_c \tag{10}$$

Where: $\bar{\varepsilon}$ is the strain at the time of damage to the specimen containing the hole and $\varepsilon_c$ denotes the strain at the time of damage to the specimen containing the hole.

The plot of creep-strain-based damage variables *versus* time can be obtained from Eq (10), and is shown in Fig 11.

The damage variable gradually increases over time for specimens with different water contents. At the same applied stress, the higher the water content of the pore-containing specimen, the greater the value of the damage variable.

The modulus of elasticity is a common method for detecting damage counts with the modulus of elasticity damage equation [33, 34]:

$$D = 1 - (E/E_0) \tag{11}$$

Where: *D* is the elastic modulus damage variable, *E* denotes the elastic modulus of the coal

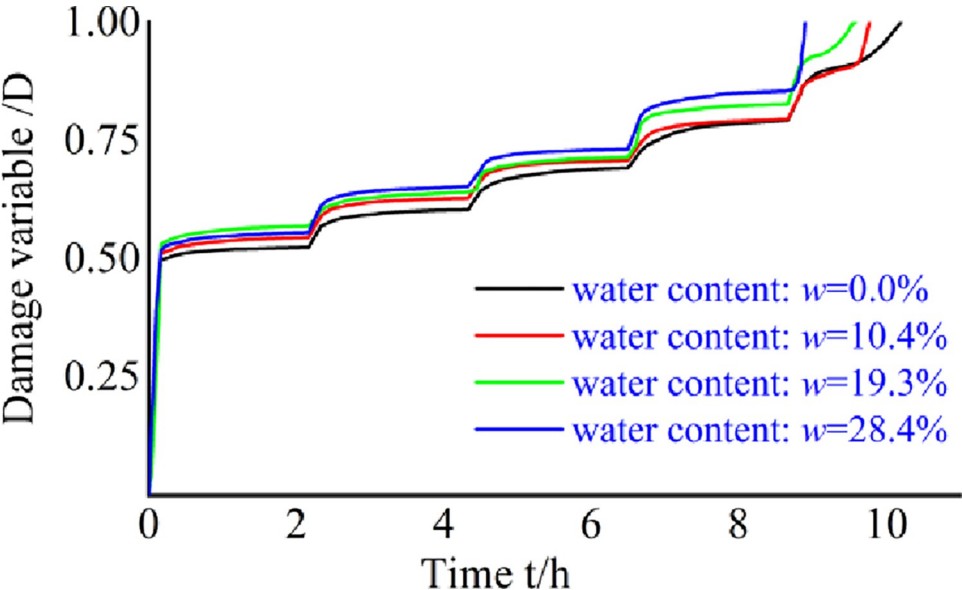

**Fig 11. Creep-strain-based damage variable curves.**

rock material at the time of damage, and $E_0$ is the elastic modulus of the coal rock material at the time of no damage.

Eq (11) can be used to plot the elastic modulus damage variables *versus* time (Fig 12).

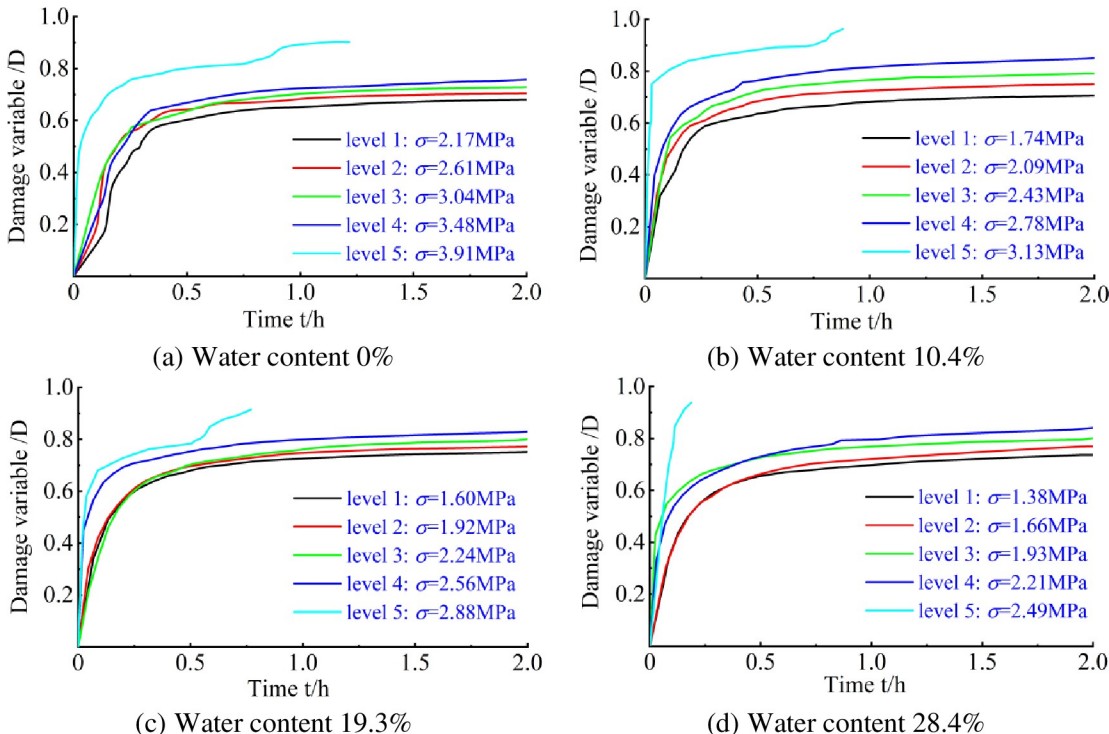

**Fig 12. Damage variable curve based on elastic modulus.** (a) Water content 0%, (b) Water content 10.4%, (c) Water content 19.3%, (d) Water content 28.4%.

As can be seen from Fig 12, with the extension of the loading time, the damage variable values for the specimens with different water contents considering the modulus of elasticity first increased rapidly and then stabilised at a certain level (between 0.6 and 0.95), with the damage variable values for the first four applied stresses ranging from 0.65 to 0.84, and the damage variable values for the higher stresses being higher than those for the lower stresses.

The reason for this is that water molecules adsorb to the surface of the coal rock material, causing a change in its surface properties. The swelling volume of the coal rock particles within the perforated specimen increases with its water content, resulting in a gradual reduction in the bond between the coal rock materials, thus reducing the strength of the coal rock materials and ultimately leading to the perforated specimen being the first to suffer damage as the applied stress and water content increase.

## Conclusions

Through self-designed experiments, the effects of different water contents on the creep process of coal rock materials were discussed, and the creep intrinsic model was improved and validated.

1. Water has a physically aggressive, softening water wedge effect on coal rock materials. Water wets the particles on the free surface of the coal body, reducing the cohesion between the particles and making the coal body around the hole more susceptible to deformation by forces. The maximum axial strain in the accelerated creep phase of the 28% water content specimen is increased by 0.31% compared to the 0% water content specimen, and its deformation between $\sigma = 50\% \sim 80\% \sigma_c$ so the stress is increased by a factor of 1.1 to 2.

2. The effect of water accelerates creep deformation in the perimeter of the boreholes, with steady-state strain increasing exponentially with applied stress and water content and is directly related to the magnitude of the water content. At each stress level, the steady-state strain is increased by 10–16% for the 10.4% water content specimen, increased by 19–22% for the 19.3% water content specimen and 31–35% for the 28.4% water content specimen compared to the steady-state strain for the 0% water content specimen.

3. On the basis of the Nishihara model, the plastic element model was introduced to establish a creep intrinsic model that considers different water-content-related damage factors. By determining the values of long-term stress $\sigma_{s1}$ and yield stress $\sigma_{s2}$ in the model and by regression analysis, it was found that the parameters of the model considering water damage were exponentially related to the water content.

4. As the water content of the coal rock material increases, the volume of the particles expands faster and their radius increases. This reduces the bonding force between the materials, which weakens the strength of the coal rock material, causing a rapid increase in the value of the damage variable with the increases in the stress level and water content of the specimens, and then stabilising to a certain level (between 0.6 and 0.95).

## Supporting information

**S1 File.**
(ZIP)

## Author Contributions

**Conceptualization:** Tianjun Zhang.

**Data curation:** Jiangbo Guo.

**Formal analysis:** Jinyu Wu.

**Methodology:** Hongyu Pan.

**Resources:** Hongyu Pan.

**Supervision:** Tianjun Zhang.

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
