## [Decision Letter · Decision Letter 0]

17 Oct 2022

PONE-D-22-26413Experimental investigation of the creep damage evolution of coal rock around gas extraction boreholes at different water contentsPLOS ONE

Dear Dr. zhang,

Thank you for submitting your manuscript to PLOS ONE. After careful consideration, we feel that it has merit but does not fully meet PLOS ONE’s publication criteria as it currently stands. Therefore, we invite you to submit a revised version of the manuscript that addresses the points raised during the review process.

Please, address all the comments made by the reviewers.

We look forward to receiving your revised manuscript.

Kind regards,

Antonio Riveiro Rodríguez, PhD

Academic Editor

PLOS ONE

2. Please note that PLOS ONE has specific guidelines on code sharing for submissions in which author-generated code underpins the findings in the manuscript. In these cases, all author-generated code must be made available without restrictions upon publication of the work. Please review our guidelines at https://journals.plos.org/plosone/s/materials-and-software-sharing#loc-sharing-code and ensure that your code is shared in a way that follows best practice and facilitates reproducibility and reuse. New software must comply with the Open Source Definition.

“The research study was carried out successfully with contribution from all authors, and all authors approved the publication of the paper. The data used to support the findings of this study are available from the corresponding author upon request. The authors declare that there is no conflict of interest regarding the publication of this paper. This work was supported by the National Natural Science Foundations of China, Study on the Research on the methane extraction synthesis technique and device innovation system in coal mine (Grant No. 51874234); Crush evolution and water-gas coupled permeability mechanism of coal body around extraction borehole (Grant No. 2021JM-390). The main research idea and manuscript preparation were contributed by Jiangbo Guo; Tianjun Zhang contributed on the manuscript preparation and performed the correlative experiment. Hongyu Pan gave several suggestions from the industrial perspectives. Jinyu Wu assisted on finalizing research work and manuscript. Finally, thanks to the test platform provided by Key Laboratory of Western Mine Exploitation and Hazard Prevention of the Ministry of Education, the test was successfully completed and the data was obtained.”

“The main research idea and manuscript preparation were contributed by Jiangbo Guo; Tianjun Zhang contributed on the manuscript preparation and performed the correlative experiment. Hongyu Pan gave several suggestions from the industrial perspectives. Jinyu Wu assisted on finalizing research work and manuscript.”

“The authors declare that there is no conflict of interest regarding the publication of this paper.”

6. We noted in your submission details that a portion of your manuscript may have been presented or published elsewhere. “This manuscript was previously submitted to a different PLOS journal as either a presubmission inquiry or a full submission.” Please clarify whether this publication was peer-reviewed and formally published. If this work was previously peer-reviewed and published, in the cover letter please provide the reason that this work does not constitute dual publication and should be included in the current manuscript.

7. We note that you have stated that you will provide repository information for your data at acceptance. Should your manuscript be accepted for publication, we will hold it until you provide the relevant accession numbers or DOIs necessary to access your data. If you wish to make changes to your Data Availability statement, please describe these changes in your cover letter and we will update your Data Availability statement to reflect the information you provide.

8. Your ethics statement should only appear in the Methods section of your manuscript. If your ethics statement is written in any section besides the Methods, please move it to the Methods section and delete it from any other section. Please ensure that your ethics statement is included in your manuscript, as the ethics statement entered into the online submission form will not be published alongside your manuscript.

Reviewers' comments:

Reviewer's Responses to Questions

**Comments to the Author**

1. Is the manuscript technically sound, and do the data support the conclusions?

Reviewer #1: Yes

Reviewer #2: Yes

Reviewer #3: Yes

2. Has the statistical analysis been performed appropriately and rigorously? 

Reviewer #1: Yes

Reviewer #2: Yes

Reviewer #3: Yes

3. Have the authors made all data underlying the findings in their manuscript fully available?

Reviewer #1: Yes

Reviewer #2: Yes

Reviewer #3: Yes

4. Is the manuscript presented in an intelligible fashion and written in standard English?

Reviewer #1: Yes

Reviewer #2: Yes

Reviewer #3: Yes

5. Review Comments to the Author

Reviewer #1: This manuscript is an informative and accessible study on the subject of creep in water-bearing coal rocks, but there is room for improvement. In my opinion, it could be a very good article with some revision and organization. To help improve the quality of this manuscript, I have made some comments below.

1. The work in this paper has some significance and value for gas extraction. Is the creep damage of the specimens applicable in the field as the coal and rock bodies around the boreholes are fractured in actual projects, whereas the specimens used in this test are not fractured?

2. The article uses the coal rock around the borehole as the research object, but the author uses a certain ratio of coal mixed with cement as the test object, rather than just coal.

3. The introductory text for the steady-state strain calculation of the specimens is missing.

4. The authors should have detailed how each of the experiments were carried out, the authors need to be more explicit in the execution of the experiments.

5. As the water content condition has a significant effect on the creep process of the coal body around the borehole, creating more fractures in its process, what is the development of fractures in the coal body during the gas extraction process? This is not shown in the text.

6. When quoting mathematical equations, please indicate the appropriate units in parentheses. For example, in equation (7), the unit of m is not indicated.

7. Conclusion 4 is too long, please rewrite it.

8. References should be up to date, the following references could be cited.

[1] Liu SM, Li XL, Wang DK. et al. Investigations on the mechanism of the microstructural evolution of different coal ranks under liquid nitrogen cold soaking, Energy Sources, Part A: Recovery, Utilization, and Environmental Effects. 2020, 1-17. https://doi.org/10.1080/15567036.2020.1841856

[2] Zhou XM, Wang S, Li XL. Research on theory and technology of floor heave control in semicoal rock roadway: Taking longhu coal mine in Qitaihe mining area as an Example. Lithosphere. 2022, 2022(11): 3810988. https://doi.org/10.2113/2022/3810988

[3] Wang, S. Li XL, Qin QZ. Study on surrounding rock control and support stability of Ultra-large height mining face. Energies. 2022, 15(18): 6811. https://doi.org/10.3390/en15186811

[4] Li XL, Chen SJ, Wang S. Study on in situ stress distribution law of the deep mine taking Linyi Mining area as an example, Advances in Materials Science and Engineering, 2021, 9(4): 5594181. https://doi.org/10.1155/2021/5594181

[5] Liu HY, Zhang BY, Li XL. Research on roof damage mechanism and control technology of gob-side entry retaining under close distance gob, Engineering Failure Analysis, 2022, 138(5), 106331. https://doi.org/10.1016/j.engfailanal.2022.106331

Reviewer #2: The creep process of the coal rock around the extraction boreholes under stress-water coupling is an important factor affecting the stability of the boreholes. To study the influence of the water content of perimeter of the coal rock around the boreholes on its creep damage, a creep intrinsic model considering water damage was established by introducing the plastic element model from the Nishihara model. To study the steadystate strain and damage evolution of coal rocks containing pores, and verify the practicality of the model, a graded loading water-bearing creep test was designed to explore the role of different water-bearing conditions in the creep process. The paper has good reference value and innovation.In the introduction part, more work should be introduced, and the corresponding analysis should be added. Such as, Creep damage model of rock mass under multi-level creep load based on spatio-temporal evolution of deformation modulus. Archives of Civil and Mechanical Engineering. 2021;21.doi:10.1007/s43452-021-00224-4. Damage evolution characteristics of saw-tooth joint under shear creep condition. International Journal of Damage Mechanics. 2021;30:453-80.doi:10.1177/1056789520974420.

Reviewer #3: COMMENT LETTER

In this paper, the creep damage evolution process of the borehole perimeter coal rock under stress-water coupling is studied more fully through a combination of theoretical and experimental methods, and a creep intrinsic model of water-bearing damage is innovatively proposed. The series of studies carried out in this paper provide a positive reflection on the deformation damage of perforated coal rocks under water-bearing conditions; however, the manuscript still suffers from the following problems.

1. The conclusions and summary need to be condensed.

2. The tests in this paper use a mixture of pulverised coal and cement as the test material, and should demonstrate that the physical and mechanical properties of the cement in the dry, watery and saturated water states are similar to those of the coal body; otherwise, the test results do not guide the "gas extraction".

3. Parameters such as modulus of elasticity are presented several times in the text and the material parameters of the specimens used in the tests are not specifically stated in detail.

4. In one stage of section 4 of the text, Figure 4.1, individual specimens in groups A, B, C and D are selected for graded creep tests and the test data are averaged to obtain axial strain curves for specimens containing pores at different moisture contents. the tests are somewhat random, does direct averaging affect the objectivity of Figure 4?

5. Is the use of steady state strain to reflect the effect of water content on the whole creep process of the specimen in the two stages in section 4 of the text unrepresentative?

6. The diagram section is not professional enough, please ask the author to adjust it.

Disposition: minor revision

6. PLOS authors have the option to publish the peer review history of their article (what does this mean?). If published, this will include your full peer review and any attached files.

Reviewer #1: No

Reviewer #2: No

Reviewer #3: No

---

## [Author Response · Author response to Decision Letter 0]

28 Oct 2022

Response to Reviewers

1. We note that several of your files are duplicated on your submission. Please remove any unnecessary or old files from your revision, and make sure that only those relevant to the current version of the manuscript are included. 

Response: OK, I have removed the duplicates and updated the file list. Please check it.

2. Thank you for stating the following financial disclosure:“This work was supported by the National Natural Science Foundations of China, Study on mechanism and parameter optimization of carbon dioxide deep hole pre-split blasting in coal seam (Grant No. 51874234).”

Response: The grant provided an important role for the thesis. I have submitted the appropriate instructions in the cover letter. Please check it.

---

## [Decision Letter · Decision Letter 1]

23 Nov 2022

Experimental investigation of the creep damage evolution of coal rock around gas extraction boreholes at different water contents

PONE-D-22-26413R1

Dear Dr. zhang,

We’re pleased to inform you that your manuscript has been judged scientifically suitable for publication and will be formally accepted for publication once it meets all outstanding technical requirements.

Kind regards,

Antonio Riveiro Rodríguez, PhD

Academic Editor

PLOS ONE

Reviewers' comments:

Reviewer's Responses to Questions

**Comments to the Author**

1. If the authors have adequately addressed your comments raised in a previous round of review and you feel that this manuscript is now acceptable for publication, you may indicate that here to bypass the “Comments to the Author” section, enter your conflict of interest statement in the “Confidential to Editor” section, and submit your "Accept" recommendation.

Reviewer #1: (No Response)

Reviewer #2: All comments have been addressed

Reviewer #3: All comments have been addressed

2. Is the manuscript technically sound, and do the data support the conclusions?

Reviewer #1: (No Response)

Reviewer #2: Yes

Reviewer #3: Yes

3. Has the statistical analysis been performed appropriately and rigorously? 

Reviewer #1: (No Response)

Reviewer #2: Yes

Reviewer #3: Yes

4. Have the authors made all data underlying the findings in their manuscript fully available?

Reviewer #1: (No Response)

Reviewer #2: Yes

Reviewer #3: Yes

5. Is the manuscript presented in an intelligible fashion and written in standard English?

Reviewer #1: (No Response)

Reviewer #2: Yes

Reviewer #3: Yes

6. Review Comments to the Author

Reviewer #1: The manuscript has been changed according to my comments, it could be accepted in the state. It is a good paper.

Reviewer #2: The authors have addressed all the comments by reviewer. i have no more comments. I think this paper can be accepted now.

Reviewer #3: (No Response)

7. PLOS authors have the option to publish the peer review history of their article (what does this mean?). If published, this will include your full peer review and any attached files.

Reviewer #1: No

Reviewer #2: No

Reviewer #3: No

---

## [Editor Report · Acceptance letter]

7 Feb 2023

PONE-D-22-26413R1 

Experimental investigation of the creep damage evolution of coal rock around gas extraction boreholes at different water contents 

Dear Dr. zhang:

I'm pleased to inform you that your manuscript has been deemed suitable for publication in PLOS ONE. Congratulations! Your manuscript is now with our production department. 

Kind regards, 

on behalf of

Dr. Antonio Riveiro Rodríguez 

Academic Editor

PLOS ONE